# Novel cell lines derived from Chinese hamster kidney tissue

**Yoshinori Kawabe[1,2], Masamichi Kamihira[1,2]***

**1** Department of Chemical Engineering, Faculty of Engineering, Kyushu University, Fukuoka, Japan,
**2** Manufacturing Technology Association of Biologics, Kobe, Japan

\* kamihira@chem-eng.kyushu-u.ac.jp

**Data Availability Statement:** Microarray data are available from the ArrayExpress database at EMBL-EBI (www.ebi.ac.uk/arrayexpress) under accession number E-MTAB-11298.

**Funding:** This work was supported in part by developing key technologies for discovering and

## Abstract

Immortalized kidney cell lines are widely used in basic and applied research such as cell permeability tests and drug screening. Although many cell lines have been established from kidney tissues, the immortalization process has not been clarified in these cell lines. In this study, we analyzed the phenotypic changes that occurred during the immortalization of kidney cells derived from Chinese hamster tissue in terms of karyotype and gene expression profiles. In the newly established cell line, designated as CHK-Q, gene expression profiles at each stage of the immortalization process and during the adaptation to serum-free conditions were analyzed by DNA microarray. Renal stem cell markers *CD24* and *CD133* were expressed in CHK-Q cells, suggesting that CHK-Q cells were transformed from renal stem cells. Kyoto encyclopedia of genes and genomes (KEGG) pathway enrichment analysis to identify the pathways of upregulated and downregulated genes revealed that the immortalization of CHK-Q cells was associated with increased fluctuations in the expression of specific proto-oncogenes. Karyotype analysis of spontaneously immortalized CHK-Q cells indicated that CHK-Q chromosomes had a typical modal number of 23 but possessed slight chromosomal abnormalities. In this study, we investigated the mechanism of cell environmental adaptation by analyzing gene expression behavior during the immortalization process and serum-free adaptation. CHK-Q cells are applicable to the fields of biotechnology and biomedical science by utilizing their characteristics as kidney-derived cells.

## Introduction

Kidney-derived cells are generally easy to culture under normal conditions and exhibit a good proliferative capability. In the long history of tissue-derived cell line establishment, many cell lines are of kidney origin [1]. Kidney-derived cell lines have been widely used in biotechnology fields such as virus production [2], vaccine production [2], recombinant protein production [2, 3], and drug screening using their membrane permeability properties [4], and have also been applied in *in vitro* studies of physiological processes such as osmoregulation and excretion [5]. In the development of renal cell lines such as 293 cells [6], BHK cells [7], and CV-1 cells [8], transformation and immortalization were achieved by viral infection and the introduction of genes encoding viral constituent proteins. Furthermore, there are many reports of

manufacturing pharmaceuticals used for next-generation treatment and diagnoses both from the Ministry of Economy, Trade and Industry, Japan (METI), and from the Japan Agency for Medical Research and Development (AMED) under grand number JP17ae0101003.

**Competing interests:** The authors have declared that no competing interests exist.

cell lines derived from diseased tissues [9] and transformed cells by overexpression of *hTERT* [10]. In addition, cells have been efficiently immortalized in combination with cell cycle regulators [11, 12]. Immortalization also alters cell properties, with changes in gene expression profiles and karyotypes [13]. Although spontaneous immortalization of renal cells such as MDCK [14] and Vero cells [15] has been reported, the mechanism of this process in such cells remains poorly understood, and their biological differences to naturally occurring cancer cells are also unclear.

The Chinese hamster ovary (CHO)-derived immortalized cells originally established by Puck et al. [16] are currently the most frequently used line for biopharmaceutical production [17]. Among them, decoding of the entire genome sequence [18, 19], omics analyses including transcriptomic [20] and proteomic [21, 22] analyses, and karyotype analysis [23] have been extensively studied in CHO-K1 cells. On the basis of this information, the development of high-quality recombinant protein production technology was performed by modifying CHO-derived immortalized cells [24, 25]. In the Manufacturing Technology Association of Biologics (MAB; cho-mab.or.jp), a Japanese Collaborative Innovation Partnership (CIP) research organization, new cell lines derived from Chinese hamsters bred as experimental animals in a controlled environment have been independently established. For example, Yamano-Adachi et al. [26] succeeded in establishing an immortalized cell line derived from Chinese hamster lung tissues, CHL-YN. CHL-YN cells show unique characteristics that differ from conventional CHO-K1 cells, such as rapid proliferation and high glutamine synthesis, considering the application of producer cell development for biopharmaceutical proteins. This result suggested that newly established cell lines might possess beneficial characteristics for basic and applied research.

In the 1950s, Puck et al. [16] examined the establishment of cell lines derived from Chinese hamsters, including CHO cells. Although they also described spontaneously immortalized cells induced from Chinese hamster kidney tissues, these cells have not been widely applied. Lavialle et al. [27] established Chinese hamster kidney-derived cell lines to investigate the changes in karyotype and chromosome number after spontaneous transformation and virus-mediated transformation using SV-40 variants, but these are not available from public cell banks. In this study, we established spontaneously immortalized cell lines from the kidney tissues of Chinese hamsters and analyzed changes in cell properties during immortalization and serum-free adaptation. We used kidney tissue from the same Chinese hamsters used for the establishment of CHL-YN cells [26] to generate cell lines from primary culture by controlling each cell passage, making it possible to track the culture history. DNA microarray analysis to follow variations in gene expression profiles was performed on the cells during the immortalization process. We finally established a new cell line derived from Chinese hamster kidney, designated as CHK-Q, and cell proliferation and gene expression were measured for cell characterization. In addition, to evaluate cell aneuploidy after immortalization, karyotype analysis was performed to investigate the chromosomal distribution of CHK-Q cells.

## Materials and methods

### Cell culture

Kidney-derived cells were cultured in DMEM/F-12 (DF) medium (042–30555; Fujifilm Wako Pure Chemical, Osaka, Japan) supplemented with 10% fetal bovine serum (FBS) (Biowest, Nuaillé, France) in 100-mm cell culture dishes coated with collagen type I (4020–010; AGC Techno Glass, Shizuoka, Japan). For cells adapted to serum-free conditions, cells were cultured in commercially available serum-free medium developed for CHO cells (31033–020, CHO-S-SFM II; Invitrogen, Waltham, MA, USA) in 100-mm cell culture dishes (BioLite 12-

556-002; Thermo Scientific, Waltham, MA, USA). For serum-free suspension culture, cells suspended in 30 mL CHO-S-SFM II medium were cultured in a 125-mL Erlenmeyer flask (431143; Corning, NY, USA) under swirling at 125 rpm in an orbital shaker (Taitec, Tokyo, Japan). For suspension culture, 1 μM 1-oleoyl lysophosphatidic acid (LPA) (62215; Cayman Chemical, Ann Arbor, MI, USA) and/or 0.1% Pluronic F-68 non-ionic surfactant (24040–032; Invitrogen) were added to the medium.

To measure cell proliferation curves, cells were seeded on 24-well tissue culture plates coated with collagen type I (AGC Techno Glass) at a cell density of $2.0 \times 10^4$ cells/well in 1.0 mL of DF medium and cultured in adherent conditions for 4 days. For suspension culture, cells were seeded in 125-mL Erlenmeyer flasks at a cell density of $2.0 \times 10^5$ cells/mL in 30 mL of CHO-S-SFMII medium and cultured under swirling for 4 days. All culture conditions were prepared in triplicate. Viable cell density was determined by the trypan blue exclusion method.

Cells were cultured at 37˚C in a 5% (v/v) $CO_2$ incubator (Panasonic Healthcare, Tokyo, Japan) and observed under a BZ-9000 digital microscope (Keyence, Tokyo, Japan).

All experiments were conducted in accordance with the relevant guidelines and regulations, and the animal experiments were approved by the Ethics Committee for Animal Experiments of the Faculty of Engineering, Kyushu University (A30-247-0).

## Cell harvesting from kidney tissues, primary and immortalized culture, and serum-free adaptation

Kidneys derived from female Chinese hamsters were obtained from a laboratory animal supplier (Charles River Laboratories Japan, Yokohama, Japan). In the supplier company, the animals were sacrificed by exsanguination under deep anesthesia before harvesting the kidneys.

Kidney tissues were minced into 1–2 mm pieces. Minced tissues were dissociated using 0.25% trypsin-EDTA solution (Invitrogen, Waltham, MA, USA) in a 37˚C water bath for 10 min, and further dispersed using an automatic tissue dissociation device (gentleMACS Tissue Dissociator; Miltenyi Biotec, North Rhine-Westphalia, Germany). After washing with DF medium containing 10% FBS, dissociated tissues and cells suspended in the medium were separately seeded into collagen-coated 100-mm dishes. Outgrowing cells were passaged when they had reached 70%–80% confluency. Then, cells were carefully passaged before reaching confluency to prevent contact inhibition. The passage culture was repeated to obtain immortalized cells (CHK-Q).

For serum-free adaptation of CHK-Q cells, serum-containing medium (DF+10% FBS) mixed with serum-free medium (CHO-S-SFM II) was used for culture. After several passages using medium mixed with a predetermined amount of serum-free medium, the proportion of serum-free medium was increased if cells actively grew in the culture conditions. This adaptation culture was repeated, and finally cells were well-adapted for serum-free conditions (CHK-Q_SF).

The limiting dilution method was separately applied to CHK-Q and CHK-Q_SF cells to obtain cell clones. Clones were selected in terms of good proliferative capacity and cell morphology.

## Karyotype analysis

Sub-confluent cells were treated with a demecolcine (Sigma–Aldrich, St Louis, MO, USA) solution at a final concentration of 0.1 μg/mL (for primary and CHK-Q cells) or 0.4 μg/mL (for CHK-Q_SF cells) for 2 h. Cells were collected after washing with PBS followed by trypsin treatment. After centrifugation, cells were suspended in 75 mM KCl solution, maintained at room temperature for 10 min, then fixed with a methanol/acetic acid (3:1) solution. Slides

with cell samples were kept warm at 45˚C or 55˚C using a block incubator (ADSTEC, Funaba-shi, Japan). Denaturation of metaphase chromosomes and hybridization using an mFISH probe (D-1526-060-DI; Metasystems, Altlussheim, Germany) were performed according to the protocols recommended by the manufacturer. Metaphase chromosomal images were captured using an Axio Imager Z2 fluorescence microscope (Carl Zeiss; Berlin, Germany) and analyzed using the ISIS software program (Metasystems). The karyotype analysis was performed by Chromocenter (Yonago, Japan).

## DNA microarray and RT-PCR analyses

Total RNA was extracted from CHK cells at each culture stage using a commercially available kit (RNeasy Mini Kit; Qiagen, Hilden, Germany). Contaminating genomic DNA was removed by DNase I (Qiagen) treatment, and the quality of total RNA was evaluated using an Agilent 2200 TapeStation (Agilent Technologies, Santa Clara, CA, USA). DNA microarray analysis for 56,105 transcript probes of 21,763 genes encoding *Cricetulus griseus* proteins was performed using commercially available DNA chips (G4858A#077089 single color 8 x 60K; Agilent Technologies). Data were quantified using Feature Extraction software (Agilent Technologies). The raw signal intensities of samples were normalized by a quantile algorithm in the preprocessCore library package on Bioconductor software [28, 29]. Microarray analysis was performed by Cell Innovator (Fukuoka, Japan). Enrichment of molecular pathways from Gene Ontology or Kyoto Encyclopedia of Genes and Genomes (KEGG) was analyzed using the DAVID (david.ncifcrf.gov) annotation tool. Heatmaps were created in the Heatmapper website (www.heatmapper.ca) [30]. Microarray data were deposited in the ArrayExpress database at EMBL-EBI (www.ebi.ac.uk/arrayexpress) under accession number E-MTAB-11298.

For RT-PCR analysis, total RNA was extracted from cells using a kit (RNAiso Plus; Takara Bio, Kusatsu, Japan), and reverse-transcribed into cDNA using reverse-transcriptase (RevaTra Ace; Toyobo, Osaka, Japan). The renal stem cell marker genes (*CD24* and *CD133*) were amplified by PCR using primer pairs shown in S1 Table. PCR was initiated with G-Taq DNA polymerase (Cosmo, Seoul, Korea) at 98˚C for 2 min, followed by 35 cycles of amplification at 98˚C for 20 s, 58˚C for 40 s, and 72˚C for 7 s, with a final extension at 72˚C for 10 min. For quantitative RT-PCR analysis, total RNA of cells was extracted and reverse-transcribed into cDNA using reverse-transcriptase and oligo-dT primers. cDNAs were mixed with corresponding primers (S1 Table) and quantitative PCR reagent (Thunderbird SYBR qPCR Mix, Toyobo) in accordance with the manufacturer's instructions. PCR was performed using a quantitative PCR device (QuantStudio3, Applied Biosystems, Waltham, MA, USA) under the following conditions: 95˚C for 1 min, followed by 45 cycles of amplification at 95˚C for 15 s, 58˚C for 15 s, and 72˚C for 30 s, with a final cycle of amplification for melting curve analysis at 95˚C for 30 s, 58˚C for 30 s, and 95˚C for 30 s. The fold change in the *CD24* and *CD133* specific transcripts relative to the glyceraldehyde-3-phosphate dehydrogenase (*GAPDH*) endogenous control gene was determined by the ΔΔCt method. The mRNA expression levels were expressed as mean values with standard deviations.

## Results

### Establishment of immortalized cells and serum-free adaptation

To track the process of spontaneous immortalization, we cultured primary cells from Chinese hamster kidney tissues, managing all cell culture histories. Chinese hamster kidneys obtained with appropriate storage and transportation after excision from the animals were minced, and the kidney tissues were enzymatically treated to dissociate cells. Primary cell culture was commenced using the outgrowth method (Fig 1A). Initially, the cells were a mixture of fibroblasts

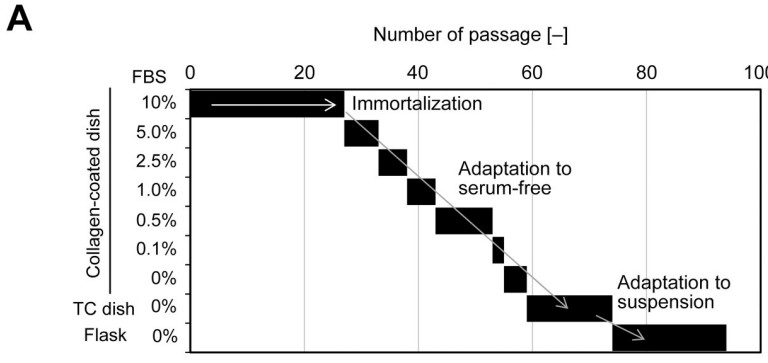

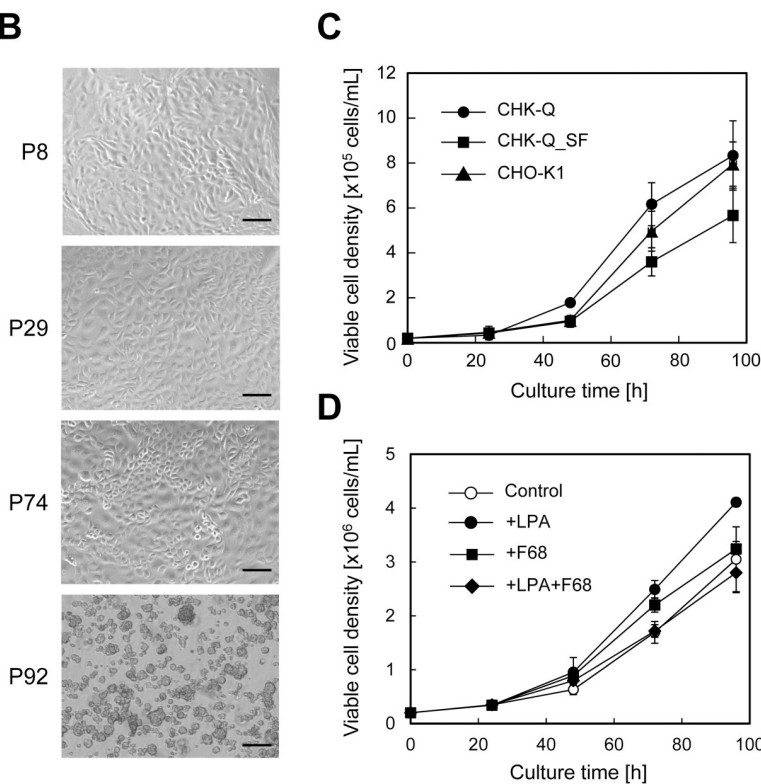

**Fig 1. Establishment of CHK-Q cells and serum-free adaptation.** (A) Process of the establishment of spontaneously immortalized cells and serum-free adaptation. (B) Cell morphology at each stage: pre-immortalized primary cells, 8th passage (P8); immortalized CHK-Q cells, 29th passage (P29); serum-free adapted CHK-Q_SF cells, 74th passage (P74); and CHK-Q_SS cells adapted to serum-free and suspension conditions, 92th (P92) passage. Scale bars = 100 μm. (C and D) Cell proliferation curve. CHK-Q, CHK-Q_SF, and CHO-K1 cells were seeded in 24-well plates containing 1 mL medium at a density of $2.5 \times 10^4$ cells/mL (C). CHK-Q_SS cells were seeded in a 125-mL Erlenmeyer flask containing 30 mL medium with 1 μM 1-oleoyl lysophosphatidic acid (LPA) and/or 0.1% Pluronic F-68 (F68) at a density of $1 \times 10^6$ cells/mL (D). Data are expressed as mean ± SD ($n = 3$).

and epithelial cells (S1 Fig). On day 31 of culture after the seventh passage, the cell population was mostly composed of epithelial cells (S1 Fig). Fig 1B shows images of the cells at each stage. Proliferation was significantly decreased at passages 8–10, but gradually accelerated thereafter. Epithelial cells were selectively cultured by adjusting the difference in adhesion to culture dishes and the timing of passaging. By repeating passaging and proliferation, cells that had spontaneously immortalized could be maintained and were completely immortalized by the

30th passage. The newly established cell line, CHK-Q, can be cryopreserved in liquid nitrogen while maintaining high viability in the same manner as general animal cells.

Next, we attempted to adapt CHK-Q cells to serum-free conditions. When cells were repeatedly cultured in a commercially available serum-free medium developed for CHO cells with stepwise decreasing serum concentrations, cells could proliferate even in low-serum and serum-free medium conditions. After several passages, cells that had adapted to serum-free conditions were obtained (CHK-Q_SF). Anchorage-free adaptation is generally achieved during serum-free adaptation, but CHK-Q_SF cells showed strong adhesion-dependent characteristics in static culture. Therefore, cells were forcibly suspended in the medium by shaking the culture for several passages, and finally adapted to serum-free and suspension conditions (CHK-Q_SS).

Subsequently, cell cloning by limiting dilution was performed for immortalized CHK-Q cells in serum-containing medium and for CHK-Q_SF in serum-free medium, and clones were selected from the viewpoints of proliferation and cell morphology. The cell proliferation curves for selected clones are shown in Fig 1C. For CHK-Q cells in serum-containing medium, the specific growth rate was 0.0472 h$^{-1}$, which was 1.2-fold higher than that of CHO-K1 cells cultured under the same conditions. The proliferation rate of CHK-Q_SF cells in serum-free medium was 0.0354 h$^{-1}$, which was lower than that of CHO-K1 cells. In the suspension culture of CHK-Q_SS cells, cell aggregation was observed (S2 Fig). To reduce cell aggregation in suspension culture, the addition of a cell aggregation inhibitor was examined. When Pluronic F-68 was added to the medium, the proliferation rate was equivalent to that of conditions without additives. In contrast, when LPA was added to the medium, cell mass formation was suppressed and the specific proliferation rate was improved to 0.0358 h$^{-1}$, which was 1.2-fold higher than that without addition (Fig 1D).

## Karyotype analysis

Karyotype abnormalities and chromosomal instability have been reported in transformed cells [2, 31]. Therefore, we investigated the karyotypes of pre-immortalized primary cells, immortalized cell clones (CHK-Q), and serum-free adapted cell clones (CHK-Q_SF) (Fig 2A and 2B). Twenty cells at each stage were examined by mFISH/FISH analysis. For the primary cells, 17 out of 20 cells showed the same normal karyotype ($2n = 22$) as wild-type Chinese hamster cells. In the remaining three cells, an increase in the number of copies of chromosome 3, a partial deletion of the long arm of the X chromosome, and a derivative chromosome from the X chromosome were observed (S3 Fig and S2 Table). Among CHK-Q cells immortalized using serum-containing medium, none of the 20 cells analyzed showed a normal karyotype, and 15 out of 20 cells showed the same karyotype. Chromosomal abnormalities of these cells included an increase in the number of copies of chromosome 10 (19/20) and a deletion of the short arm of chromosome 9 (20/20) (S4 Fig and S3 Table). For serum-free-adapted CHK-Q_SF cells, none of the 20 cells analyzed showed the original normal karyotype, and 8 out of 20 cells showed the same karyotype. Increases in the number of copies of chromosome 10 (20/20), hybrid chromosomes consisting of a partial overlap of the short-arm region of chromosome 2 and regions derived from chromosomes 2 and 4 (19/20), regions derived from chromosomes 3 and 4 (18/20), and regions derived from chromosomes 6 and 9 (20/20) were observed (S5 Fig and S4 Table).

The chromosome number (modal number) in primary cells was mostly $2n = 22$ (95%; 19/20), the same as that for Chinese hamster chromosomes [18, 19], whereas the modal number was $2n = 23$ (80%; 16/20) in immortalized cells (CHK-Q), and $2n = 24$ (70%; 14/20) in serum-free-adapted cells (CHK-Q_SF) (Fig 2B). It was found that the increased number of

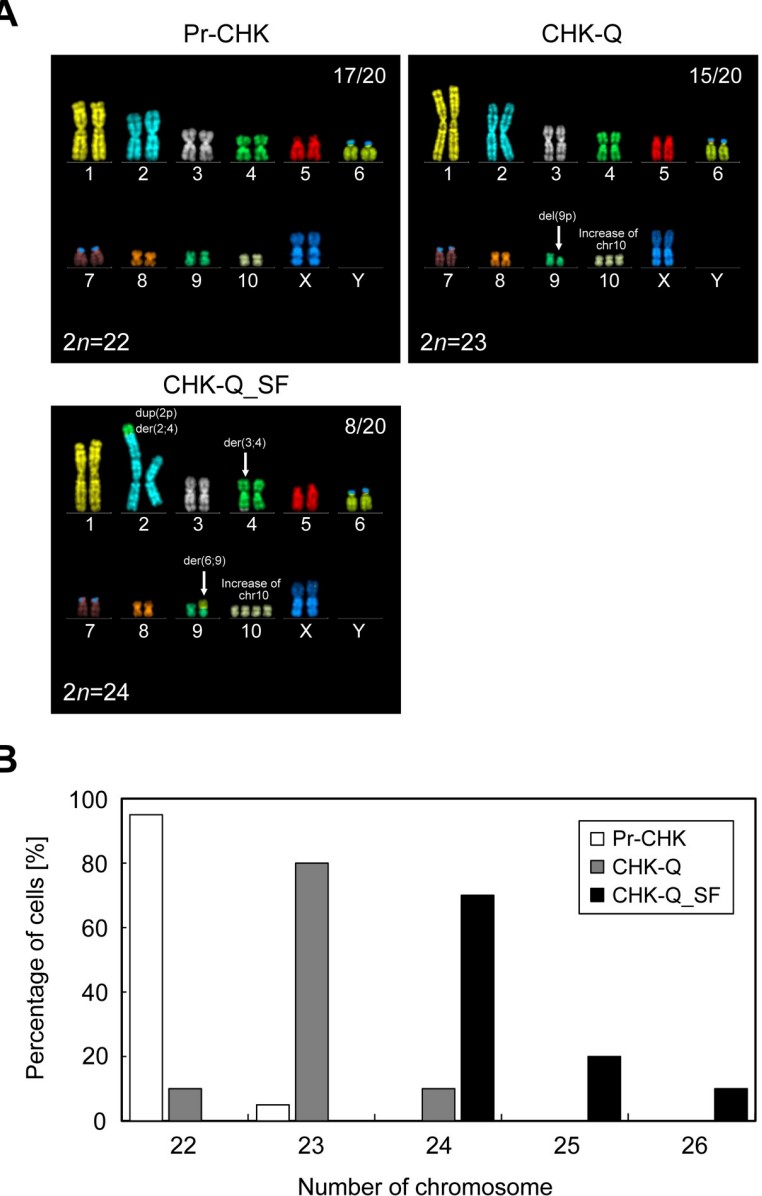

**Fig 2. Karyotype analysis and chromosome distribution.** (A) Chromosomal structure of Chinese hamster kidney-derived primary cells, Pr-CHK, immortalized CHK-Q, and serum-free-adapted CHK-Q_SF cells. Cells were subjected to multicolor FISH/FISH analysis to evaluate the karyotype. All results of the karyotype analysis are summarized in S2–S4 Tables. del(9p), deletion of the short arm of chromosome 9; der(2;4), derivative chromosome containing chromosome 2- and 4-derived regions; dup(2p), duplication of the short arm of chromosome 2, der(3;4), derivative chromosome containing chromosome 3- and 4-derived regions; der(6;9), derivative chromosome containing chromosome 6- and 9-derived regions. (B) Distribution of chromosome numbers in Pr-CHK, CHK-Q and CHK-Q_SF cells. The number of chromosomes was measured in 20 cells at each passage.

chromosomes was due to increases in chromosome 10. In CHK-Q cells, in addition to the increases in chromosome 10, a short arm defect on chromosome 9 was observed. In CHK-Q_SF cells, translocations between chromosome numbers 2 and 4, 3 and 4, and 6 and 9 were observed. No deletions, derivations, or duplications were identified on chromosomes 1, 5, 7, and 8 in cells throughout the establishment period.

## Gene expression analysis using DNA microarray

The entire gene transcripts were analyzed by DNA microarray to evaluate the overall behavior of gene expression in primary cells, immortalized CHK-Q cells, serum-free-adapted CHK-Q_SF cells, and suspension-adapted CHK-Q_SS cells. Principal component analysis (PCA) showed that the primary cells were different to CHK-Q cells and completely different to the CHO-K1 cells used as a control (Fig 3A). A similar tendency was observed in the heat map analysis (Fig 3B). When a heat map was created for kidney-related genes, different expression fluctuations were confirmed at each cell stage (Fig 3C). The expression of *CD24* and *CD133* [32], which are renal stem cell markers, was increased after immortalization (S6 Fig). By quantitative RT-PCR analysis, the expression levels of *CD24* and *CD133* in immortalized CHK-Q cells were increased by at least 4- and 19-fold, respectively, compared with those in primary cells (Fig 3D). The elevated expression of other renal stem cell marker genes such as paired box 2 (*PAX2*), paired box 8 (*PAX8*), homeobox B7 (*HOXB7*) and CBP/p300 interacting transactivator with Glu/Asp rich carboxy-terminal domain 1 (*CITED1*) [33, 34] was also detected in immortalized CHK cells (S6 Fig). However, the expression of renal stem cell markers such as cadherin 11 (*CDH11*), eyes absent homolog 1 (*EYA1*), and sineoculis homeobox homolog 1 (*SIX1*) [33, 34] was detected in primary cells but subsequently declined (S6 Fig). Considering these results, we concluded that CHK-Q cells might be transformed from renal stem cells.

Fig 3E shows the expression behavior of genes encoded on chromosome 10, the number of copies of which was increased in CHK-Q cells. Carboxypeptidase (*CPQ*), which was highly expressed in primary cells, and collagen triple helix repeat containing 1 (*CTHRC1*) and hyaluronan synthase 2 (*HAS2*), which are genes related to cell adhesion and migration, were gradually suppressed during the passages. In contrast, the expression levels of antioxidant gene oxidation resistance 1 (*OXR1*), protein kinase gene tribbles pseudokinase 1 (*TRIB1*), and ubiquitin protein ligase E3 component n-recognin 5 (*UBR5*), which is a regulator of the DNA damage response, were increased by more than 2-fold in CHK-Q cells compared with primary cells.

## Gene expression analysis during the immortalization process

In general, cells are immortalized through the regulation of genes involved in processed such as the cell cycle, p53/pRB pathway, MAPK pathway, and oxidative stress pathway [35]. To investigate the mechanism associated with the immortalization of CHK-Q cells, a comprehensive gene expression analysis was performed during the immortalization process from pre-immortalized primary culture (Fig 4A). Gene transcripts of cells at each passage during culture, from primary cells after passage 10 of cells outgrowing from kidney tissues (P10), 15 and 22 passages after the beginning of primary culture (P15 and P22), and immortalized CHK-Q cells at passage 29 (P29), were subjected to DNA microarray analysis. For each cell lines from P10, P15, P22, and P29, many genes in which the expression levels fluctuated by 2-fold or more and 0.5-fold or less were observed in the heat map as compared with primary cells (Fig 4A). The extracted genes did not belong to a specific gene cluster, but the expression levels were changed in a wide range of genes. To further narrow down the variable genes, those with a *Z* score of +2 or more (upregulation) and −2 or less (downregulation) were extracted as expression-variable genes (Fig 4B). Upregulation of 807 genes at P10 was observed as compared with primary cells, and the number was decreased as the passage number progressed to P15, P22, and P29 (536, 422, and 334 genes, respectively). In contrast, the number of downregulated genes (925 at P10 as compared with primary cells) was increased with the progression of passage number (594, 611, and 706 genes for P15, P22, and P29, respectively).

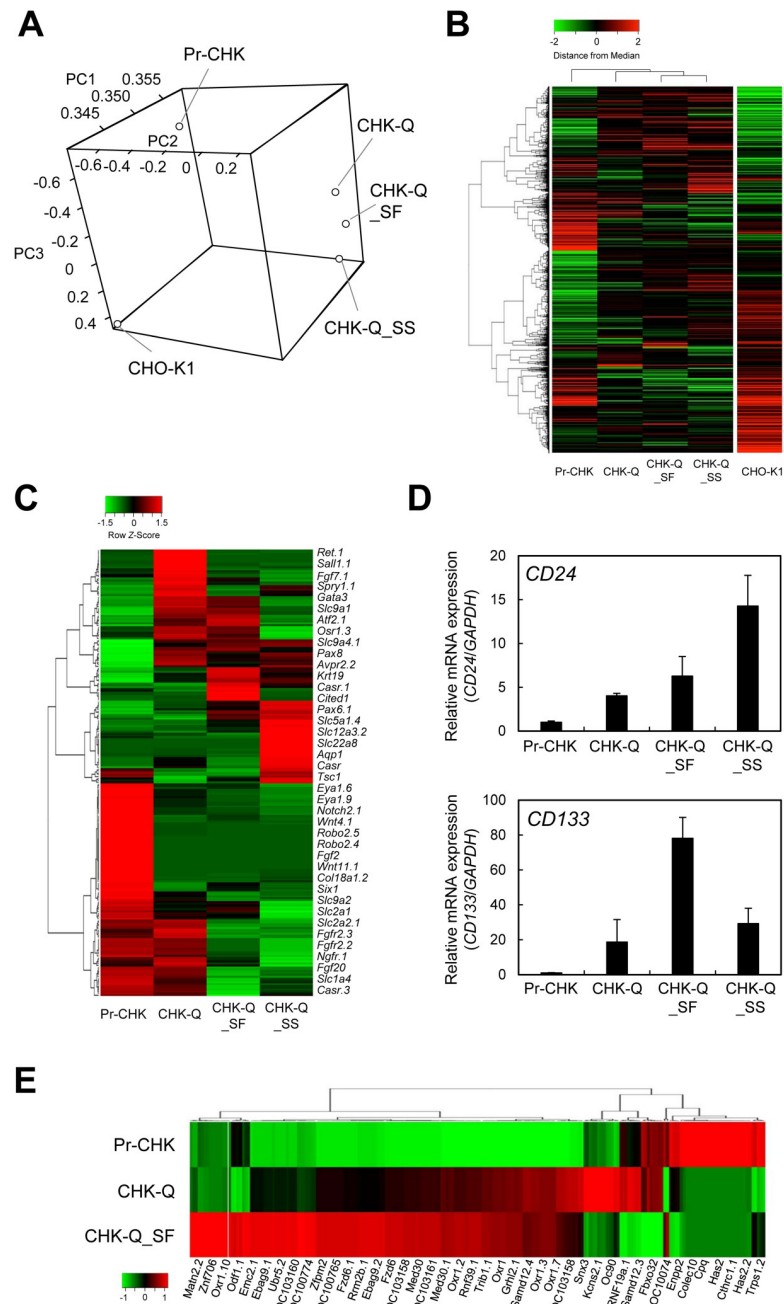

**Fig 3. Comparative gene expression analysis using DNA microarrays.** (A) Three-dimensional principal component analysis (PCA) for transcriptomic data of pre-immortalized primary cells (Pr-CHK), immortalized cells (CHK-Q), serum-free-adapted cells (CHK-Q_SF), CHK-Q_SS cells that were adapted for serum-free and suspension conditions, and CHO-K1 cells. (B) Heatmap analysis of expression profiles of global genes. Hierarchical clustering was analyzed using the Pearson's correlation distance/average linkage method. (C) Heatmap analysis of expression profiles of kidney-related genes. (D) Expression analysis of kidney stem cell markers (*CD24* and *CD133*) by quantitative RT-PCR. Data are expressed as mean ± SD (*n* = 3). (E) Heatmap analysis of fluctuating genes expressed on chromosome 10 of Chinese hamster.

To analyze the overall behavior of the extracted genes regarding metabolic and signal transduction pathways, the biological importance was interpreted by KEGG pathway analysis. Fig 4C shows the pathways that exhibited a significant difference in the enrichment analysis. For

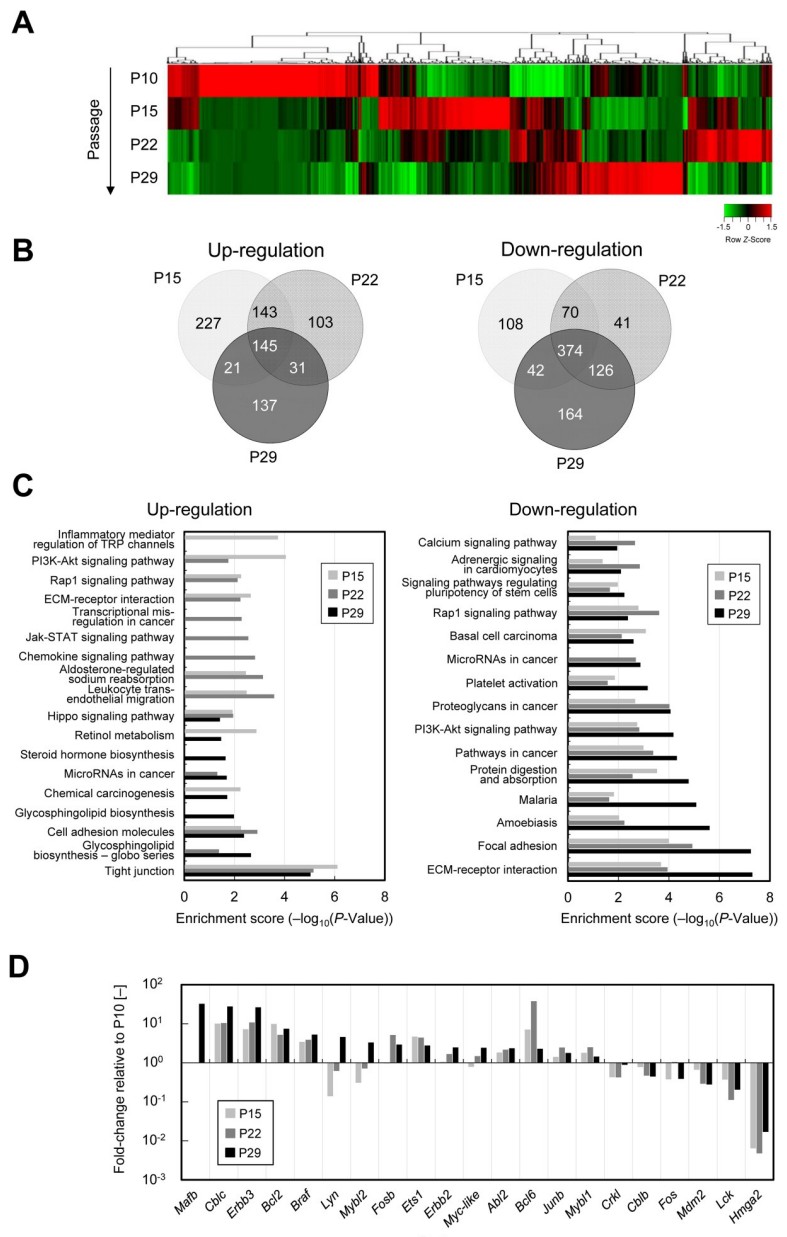

**Fig 4. Gene expression analysis during the spontaneous immortalization process.** (A) Heatmap analysis of gene expression in CHK cells during the immortalization process. Gene transcripts were prepared from CHK cells during culture at passages 10 (P10), 15 (P15), 22 (P22), and 29 (P29) after starting the primary culture. (B) Venn diagrams based on DNA microarray analysis. Differentially expressed genes were extracted according to changes of 2-fold or more and 0.5-fold or less, and a *Z* score of +2 or more and −2 or less. The numbers shown in the Venn diagram indicate significantly upregulated and downregulated gene numbers compared with P10. (C) KEGG pathway analysis associated with the immortalization process. Enrichment scores (−log$_{10}$(*P*-value)) for significant pathways in upregulated and downregulated genes extracted in (B) at P15, P22, and P29 compared with those of P10. (D) Differentially expressed genes of proto-oncogenes at P15, P22, and P29 in comparison to P10.

upregulated genes, inflammatory mediator regulation and PI3K–AKT signaling of TRP channel were extracted for P15, while the JAK–STAT and chemokine signaling pathways were extracted for P22, and the glycosphingolipid biosynthesis pathway was extracted for P29. Throughout the immortalization process, consistently high expression of tight junction-related

genes such as Claudin (*CLUD1*, *CLUD3*), Occludin (*OCLN*), and tight junction protein 3 (*TJP3*) was observed (S7 Fig). For downregulated genes, a very significant decrease in the expression of adhesive genes such as extracellular matrix (ECM) receptor interaction and focal adhesion was observed. From these results, the expression-variable genes with a significant difference compared with P10 showed a clear difference in their upregulation at each analysis point. In contrast, the downregulated genes fluctuated throughout the immortalization process. Cell cycle- and aging-related genes were also investigated. However, no significant fluctuations were observed (S8 Fig). Next, the expression of proto-oncogenes that have the potential to induce cancer were investigated (Fig 4D). Proto-oncogenes v-maf avian musculoaponeurotic fibrosarcoma oncogene homolog B (*MAFB*), Cbl proto-oncogene C (*CBLC*), v-erb-b2 avian erythroblastic leukemia viral oncogene homolog 2 (*ERBB2*), v-erb-b2 avian erythroblastic leukemia oncogene homolog 3 (*ERBB3*), B-cell CLL/lymphoma 2 (*BCL2*), B-cell CLL/lymphoma 6 (*BCL6*), v-raf murine sarcoma viral oncogene homolog B (*BRAF*), FBJ murine osteosarcoma viral oncogene homolog B (*FOSB*), v-ets avian erythroblastosis virus E26 oncogene homolog 1 (*EST1*), v-myb avian myeloblastosis viral oncogene homolog-like 1 (*MYBL1*), v-myb avian myeloblastosis viral oncogene homolog-like 2 (*MYBL2*), myc proto-oncogene protein (*MYC*), c-abl oncogene 2, non-receptor tyrosine kinase (*ABL2*), and Jun B proto-oncogene (*JUNB*), were expressed by more than 2-fold in immortalized cells compared with primary cells. Furthermore, significant suppression of the expression of high mobility group AT-hook 2 (*HMGA2*), which is an epithelial–mesenchymal transition molecule [36], was observed, indicating the epithelial character of CHK-Q cells.

## Discussion

In this study, we established a novel cell line, CHK-Q, that was spontaneously immortalized from primary cells derived from Chinese hamster kidney tissues. Although previous studies have revealed the mechanism of spontaneous immortalization in the process of cell line establishment, no studies have been supported by sufficient data such as a comprehensive analysis of gene transcripts. Therefore, we investigated the changes in gene expression profiles of cells during the immortalization process from primary culture by DNA microarray analysis. Compared with primary cells at P10, the number of upregulated expression-variable genes were decreased, and those that were downregulated were increased as the immortalization process progressed to P29. During the immortalization process, the cell population tended to have a more uniform gene expression profile, with wider differences and reduced diversity compared with pre-immortalized cells, indicating qualitative changes in the cells. In the early stages of culture (P15), upregulation of genes related to inflammatory mediator regulation of TRP channels was observed. TRP channels have been shown to be associated with cell proliferation [37]. It was suggested that the activation of $Ca^{2+}$ influx during this period may be involved in the proliferation response of CHK cells. After achieving immortalization (P29), the expression of genes associated with the glycosphingolipid biosynthesis pathway was enhanced. The upregulation of glycosphingolipid biosynthetic pathway-related genes has been shown to be associated with the promotion of cell proliferation [38, 39]. Thus, it is considered that upregulation of these genes may be involved in the immortalization of CHK cells. In addition, high expression of tight junction-related genes such as *TJP3* and *CLDN1* was consistently observed during the immortalization process. Both forced expression and knockdown of these genes are known to be associated with cell proliferation and carcinogenesis [40]. Because tight junction-related genes are strongly expressed in CHK-Q cells, these cells can be used for pharmacological research such as drug transport and reabsorption evaluation, and for the development of absorption promotion technology mediated by tight junction control. The pathways extracted

from downregulated genes were frequently identified throughout the immortalization process. In particular, there was a significant decrease in the expression of adhesion-related genes such as ECM receptor and focal adhesion. There are few reports on the relationship between decreased expression of adhesion-related genes and immortalization, and further research may be needed.

The telomere enzyme TERT, cell cycle p53/pRb, and methyltransferase DNMT1 have been reported to be involved in cellular senescence and immortality [35, 41]. However, no significant changes in the expression of these genes were observed during the immortalization process. In contrast, the expression of proto-oncogenic genes *MAFB*, *LYN*, *MYBL2*, and *MYC* were increased in the latter period of the immortalization process by 32-, 4.5-, 3.3-, and 2.4-fold, respectively, at P29 compared with that at P10. The expression of *CBLC*, *ERBB3*, and *BRAF* at P29 and *BCL6* at P22 was observed to be 10-fold higher than that in primary cells during the culture period. In the case of *CBLC* and *ERBB3*, 5-fold higher expression was consistently confirmed during the immortalization process. These proto-oncogenes have been reported to be strongly associated with cell proliferation in studies demonstrating cell proliferation upregulation by forced expression of *MAFB* [42], tumor cell proliferation by heterodimerization with *ERBB2* and *ERBB3* [43], expression of *BCL2* and *BCL6* involved in anti-apoptotic activity [44, 45], and cancer cell proliferation inhibitory effects using shRNAi against *CBLC* [46]. A serine/threonine protein kinase, *BRAF* is also one of the most well-known oncogenes [47]. Therefore, it was suggested that fluctuations in the expression of these genes affect the immortalization process.

Karyotype abnormalities and chromosomal instability are often observed in transformed cells [31, 48]. In this study, we evaluated the karyotype of chromosomes in primary cells as well as in CHK-Q cells. The normal karyotype of Chinese hamsters is comprised of 10 autosomal chromosomes and 2 sex chromosomes, totaling 22 chromosomes in a diploid cell [18, 19]. Among the analyzed cells, the modal numbers were 22 to 24. Chromosomal increases were observed in CHK-Q and CHK-Q_SF cells and were due to the increase of chromosome 10. Among genes encoded on chromosome 10, high expression fluctuation of the antioxidant-related gene *OXR1* was observed in CHK-Q cells and serum-free-adapted CHK-Q_SF cells. *OXR1* has been reported to be involved in the cell cycle and in genomic stability maintenance [49, 50]. Although further investigations are required, it was suggested that the increased number of copies of chromosome 10 may be associated with the immortalization and genomic stability of CHK cells in response to changes in the culture environment. In addition to chromosomes 1 and 8, which are known to be stable in Chinese hamster ovary (CHO) cells [23], chromosomes 5 and 7 also remained intact in CHK cells. These results indicate that these chromosomes may be resistant to rearrangement.

In CHK cells, co-expression of renal stem/progenitor cell markers *CD24* and *CD133* [32] was detected by microarray and RT-PCR analyses (S9 Fig), while only CD24 was expressed in HEK293 cells [51]. This suggests that CHK-Q cells may possess the properties of renal stem/progenitor cells. When the culture conditions of CHK-Q cells are well controlled for differentiation, functional renal expression may be elicited. We are currently conducting research to create a renal model using CHK-Q cells.

In conclusion, we established spontaneously immortalized CHK-Q cells from Chinese hamster kidney tissues through repeated passages from primary culture. Analysis of the karyotype of CHK cells revealed that they underwent minimum rearrangement throughout the culture period and retained a near-normal karyotype. To analyze the immortalization process of CHK cells, a comparison of gene expression fluctuations with primary cells was performed using DNA microarrays, and KEGG pathway enrichment analysis was performed to identify upregulated and downregulated pathways. The immortalization might be caused by increased

gene expression fluctuation and activation of proto-oncogenes, suggesting that CHK cells are derived from renal stem cells. Using the characteristics of renal stem cell origin, CHK-Q cells may be a new cell resource for constructing renal models. Furthermore, the cell line could be used as a host for the production of biopharmaceuticals, alongside CHO cells and other kidney-derived cell lines.

## Supporting information

**S1 Table. Primers used for RT-PCR.**
(DOCX)

**S2 Table. Karyotype analysis of pre-immortalized primary cells.**
(DOCX)

**S3 Table. Karyotype analysis of immortalized CHK-Q cells.**
(DOCX)

**S4 Table. Karyotype analysis of serum-free-adapted CHK-Q_SF cells.**
(DOCX)

**S1 Fig. Cell morphology at 1st and 7th passages (P1 and P7).** Scale bars = 200 μm.
(TIF)

**S2 Fig. Cell morphology of suspension culture of CHK-Q_SS cells using cell aggregation inhibitors (on day 3).** Cells were cultured in the absence (Control) or presence of 0.1% Pluronic F-68 (F68) and/or 1 μM 1-oleoyl lysophosphatidic acid (LPA). Scale bars = 100 μm.
(TIF)

**S3 Fig. Karyotype analysis (mFISH/FISH images) of pre-immortalized primary cells (minor karyotypes).** del(Xq), deletion of the long arm of X chromosome; der(X), derivative chromosome containing X chromosome-derived regions.
(TIF)

**S4 Fig. Karyotype analysis (mFISH/FISH images) of immortalized CHK-Q cells (minor karyotypes).** del(9p), deletion of the short arm of chromosome 9; del(Xq) and del(Xp), deletion of the long and short arms of the X chromosome, respectively.
(TIF)

**S5 Fig. Karyotype analysis (mFISH/FISH images) of serum-free-adapted CHK-Q_SF cells (minor karyotypes).** dup(3q), duplication of the long arm of chromosome 3; t(4p7q), translocation between the short arm of chromosome 4 and the long arm of chromosome 7, der(X;10), derivative chromosome containing X chromosome- and chromosome 10-derived regions; der(3;9), derivative chromosome containing chromosome 3- and 9-derived regions; der(X;3;4), derivative chromosome containing X chromosome- and chromosome 3- and 4-derived regions; der(3;5), derivative chromosome containing chromosome 3- and 5-derived regions; ins(2;4), insertion of chromosome 4-derived regions into chromosome 2.
(TIF)

**S6 Fig. Expression analysis of renal stem cell marker genes using DNA microarray.** Relative gene expression of renal stem cell markers (*CD24*, *CD133*, *PAX2*, *PAX8*, *HOXB7*, *CITED1*, *EYA1*, *SIX1*, and *CDH11*) for Pr-CHK, CHK-Q, CHK-Q_SF and CHK-Q_SS cells.
(TIF)

**S7 Fig. Expression analysis of tight junction-related genes using DNA microarray.** Relative gene expression of tight junction-related genes (*CLUD1*, *CLUD3*, *OCLN*, and *TJP3*) in CHK

cells during culture at passages 10 (P10), 15 (P15), 22 (P22), and 29 (P29).
(TIF)

**S8 Fig. Expression analysis of senescence and cell cycle-related genes using DNA microarray.** Relative gene expression of representative senescence and cell cycle-related genes (*TERT*, *RB*, *CDK1*, and *CDCK25A*) in CHK cells during culture at passages 10 (P10), 15 (P15), 22 (P22), and 29 (P29).
(TIF)

**S9 Fig. RT-PCR analysis of renal stem cell markers.** The renal stem cell marker genes (*CD24* and *CD133*) were amplified by PCR using the primer pairs shown in S1 Table. Lane M, DNA molecular weight markers (phiX174–*Hin*cII digest); lane 1, kidney tissue cells of Chinese hamster; lane 2, pre-immortalized primary cells; lanes 3–4, two CHK-Q cell clones; lane 5, CHK-Q_SF cell clone; lane 6, CHO-K1 cells; lane W, distilled water. *GAPDH* was used as an internal control.
(TIF)

**S1 Raw images.**
(PDF)

## Acknowledgments

The authors wish to thank Mr. Sho Fujiwara, Ms. Hiroko Yamanaka, and Ms. Yuuki Amamoto for their technical assistance, and Dr. Koichi Nonaka and Dr. Takeshi Omasa for their advice on the procedures. We also thank H. Nikki March, PhD, from Edanz (https://jp.edanz.com/ac) for editing a draft of this manuscript.

## Author Contributions

**Conceptualization:** Masamichi Kamihira.

**Data curation:** Yoshinori Kawabe, Masamichi Kamihira.

**Formal analysis:** Yoshinori Kawabe, Masamichi Kamihira.

**Funding acquisition:** Masamichi Kamihira.

**Investigation:** Yoshinori Kawabe, Masamichi Kamihira.

**Methodology:** Masamichi Kamihira.

**Project administration:** Masamichi Kamihira.

**Resources:** Masamichi Kamihira.

**Supervision:** Masamichi Kamihira.

**Validation:** Yoshinori Kawabe, Masamichi Kamihira.

**Visualization:** Yoshinori Kawabe, Masamichi Kamihira.

**Writing – original draft:** Yoshinori Kawabe, Masamichi Kamihira.

**Writing – review & editing:** Yoshinori Kawabe, Masamichi Kamihira.

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
