## [Decision Letter · Decision Letter 0]

28 Jan 2022

PONE-D-22-00223Novel cell lines derived from Chinese hamster kidney tissuePLOS ONE

Dear Dr. Kamihira,

Thank you for submitting your manuscript to PLOS ONE. After careful consideration, we feel that it has merit but does not fully meet PLOS ONE’s publication criteria as it currently stands. Therefore, we invite you to submit a revised version of the manuscript that addresses the points raised during the review process.

ACADEMIC EDITOR: Please include additional data to characterize the cell line, and add more information about methodology as requested by the reviewers. ==============================

We look forward to receiving your revised manuscript.

Kind regards,

Wuqiang Zhu, MD, PhD

Academic Editor

PLOS ONE

Journal Requirements:

Reviewers' comments:

Reviewer's Responses to Questions

**Comments to the Author**

1. Is the manuscript technically sound, and do the data support the conclusions?

Reviewer #1: Partly

Reviewer #2: Yes

2. Has the statistical analysis been performed appropriately and rigorously? 

Reviewer #1: I Don't Know

Reviewer #2: Yes

3. Have the authors made all data underlying the findings in their manuscript fully available?

Reviewer #1: No

Reviewer #2: Yes

4. Is the manuscript presented in an intelligible fashion and written in standard English?

Reviewer #1: Yes

Reviewer #2: Yes

5. Review Comments to the Author

Reviewer #1: • In the chapter of result, the authors referred “Initially, the cells were a mixture of fibroblasts and epithelial cells (S1 Fig). On day 31 of culture after the seventh passage, the cell population was mostly composed of epithelial cells (S1 Fig).” The author just shows the bright field pictures of the cells, and does not prove the properties of the cells through some specific markers. If some immunofluorescence staining pictures of fibroblast markers such as alpha Smooth Muscle Actin and epithelial cell markers such as cytokeratins or E-cadherin can be provided it will be more convincing.

• How the authors got the cell proliferation curve of FIGURE 1C and 1D? The authors may have briefly described the cell seeding method for this experiment. But I did not find information about data collection. I think it should be stated in the chapter of MATERIALS AND METHODS.

• The authors referred “the expression of CD24 and CD133 [30], which are renal stem cell markers, was increased, and CD24 expression was maintained at a high level even after immortalization (Fig 3D).”, About the expression of CD24 and CD133 it would be more intuitive and convincing if the authors could add immunofluorescence staining experiments, the bar graphs of qPCR and western blot can be more intuitive and convincing.

• The authors referred “The immortalization might be caused by increased gene expression fluctuation and activation of proto-oncogenes, suggesting that CHK cells are derived from renal stem cells.“ The analysis of the causes of spontaneous immortalization in this cell line is overly dependent on the results of DNA microarray. If the authors can proceed further study about the relation between proto-oncogenes and cell immortalization I believe the article will be very interesting.

Reviewer #2: Immortalized cell lines exhibit a high level of genetic and phenotypic diversity and instability. In the current study, the authors established a new immortalized kidney cell lines from Chinese hamster tissue. They analyzed the phenotypic changes that occurred during the immortalization of kidney cells derived from Chinese hamster tissue in terms of karyotype and gene expression profiles. The paper is well written. The Introduction and Discussion sections provide useful information for the readers. It seems to me an original work and of high applicability.

There are several issues the authors need to address to improve the manuscript.

1. The authors detect the expression of CD24 and CD133, which are renal stem cell markers. Did the authors detect other renal stem cell marker genes? There is not enough evidence to suggest CHK-Q cells might be transformed from renal stem cells.

2. The authors established a new immortalized kidney cell lines from Chinese hamster tissue. What is the advantage of these new immortalized kidney cell lines? What is the application of CHK-Q cells?

3. The chromosomes bear structural abnormality and undergo changes in structure and number during cell proliferation in CHK-Q cells. The cell line is unstable and may lose productivity over time in the manufacturing process. In the manuscript, the authors suggest the cell line could be used as a host for the production of biopharmaceuticals, alongside CHO cells and other kidney-derived cell lines. Did the authors detect productivity (for example: lentivirus package) over time in the lab?

4. In lines 150-151: “Data were quantified using Feature Extraction software (Agilent Technologies) and normalized using R statistical processing software.” How do the authors normalize the DNA microarray data? Is raw intensity data for each experiment log10 transformed and then used for the calculation of Z scores?

5. In lines 263-265: “(PCA) showed that the primary cells were different to CHK-Q cells and completely different to the CHO-K1 cells used as a control (Fig 3A). A similar tendency was observed in the heat map analysis (Fig 3B).” However, it is not easy to find this tendency in Fig 3B.

6. These figures are low-resolution images. Figure legends maybe can separately from Results part on a page, at the end of the manuscript.

6. PLOS authors have the option to publish the peer review history of their article (what does this mean?). If published, this will include your full peer review and any attached files.

Reviewer #1: No

Reviewer #2: No

---

## [Author Response · Author response to Decision Letter 0]

9 Mar 2022

Response to Academic Editor:

Comment:

Please include additional data to characterize the cell line, and add more information about methodology as requested by the reviewers.

Response:

We have addressed all the comments raised by the reviewers. We hope that our explanations and revisions are satisfactory. In addition, we re-confirmed the journal requirements: 1) PLOS ONE’s style requirements, 2) the experiments involving animals and 3) data not shown. 

In the experiments involving animals, we have not bred animals and kidney tissues were purchased from the laboratory animal supplier (Charles River Laboratories, Japan). According to the supplier company, female Chinese hamsters were sacrificed by exsanguination under deep anesthesia before harvesting the kidneys, and the animal experiment was conducted under the ethical principles and guidelines in the company. The sentences have been included in the manuscript (Page 8, Line 116).

We removed the “data not shown” phrases and provided the related results as supplementary figures (S6-S8 Figs).

To Reviewer 1:

Thank you very much for the valuable comments that helped us to improve our paper. According to your comments, we revised the manuscript.

Comment 1:

In the chapter of result, the authors referred “Initially, the cells were a mixture of fibroblasts and epithelial cells (S1 Fig). On day 31 of culture after the seventh passage, the cell population was mostly composed of epithelial cells (S1 Fig).” The author just shows the bright field pictures of the cells, and does not prove the properties of the cells through some specific markers. If some immunofluorescence staining pictures of fibroblast markers such as alpha Smooth Muscle Actin and epithelial cell markers such as cytokeratins or E-cadherin can be provided it will be more convincing.

Response 1:

Thank you for the comment. Since it is difficult to obtain anti-Chinese hamster antibodies for specific marker proteins, commercially available anti-mouse antibodies (alpha smooth muscle actin [#ab7817, Abcam] and cytokeratin 19 [#ab220193, Abcam]) were used for immunostaining. Unfortunately, however, these antibodies did not react with Chinese hamster-derived proteins. 

The heterogeneity of cells during early stage of passage culture was also confirmed by RT-PCR of marker genes specific for epithelial cells and fibroblasts, as shown in Figure A. Fibroblast marker genes (Collagen 1 A1, aSMA and Vinentin) and epithelial marker genes (E-cadherin, Cytokeratin19 and EpCAM) were clearly detected in cells from the second passage, while fibroblast marker genes of Collagen 1 A1 and aSMA were not detected in cells from the ninth passage. 

Figure A. RT-PCR analysis for fibroblast and epithelial marker genes. Lane M, phiX-174-HincII digest; Lane 1, distilled water; Lane 2, 2nd passage; Lane 3, 9th passage. Glyceraldehyde-3-phosphate dehydrogenase (GAPDH) gene was used as an internal control.

Comment 2:

How the authors got the cell proliferation curve of FIGURE 1C and 1D? The authors may have briefly described the cell seeding method for this experiment. But I did not find information about data collection. I think it should be stated in the chapter of MATERIALS AND METHODS.

Response 2:

According to your suggestion, we added the following sentences in Page 7, Line 101.

(Page 7, Line 101)

To measure cell proliferation curves, cells were seeded on 24-well tissue culture plates coated with collagen type I (AGC Techno Glass) at a cell density of 2.0 × 104 cells/well in 1.0 mL of DF medium and cultured in adherent conditions for 4 days. For suspension culture, cells were seeded in 125-mL Erlenmeyer flasks at a cell density of 2.0 × 105 cells/mL in 30 mL of CHO-S-SFMII medium and cultured under swirling for 4 days. All culture conditions were prepared in triplicate. Viable cell density was determined by the trypan blue exclusion method.

Comment 3:

The authors referred “the expression of CD24 and CD133 [30], which are renal stem cell markers, was increased, and CD24 expression was maintained at a high level even after immortalization (Fig 3D).”, About the expression of CD24 and CD133 it would be more intuitive and convincing if the authors could add immunofluorescence staining experiments, the bar graphs of qPCR and western blot can be more intuitive and convincing.

Response 3:

We have tried immunostaining of CD24 and CD133 on CHK cells using commercially available anti-mouse CD24 antibodies (#MA5-11828, Invitrogen; #130-110-825, Miltenyi Biotec) and anti-rat or rabbit CD133 antibodies (#14-1331-82, Invitrogen or #NB120-16518, Miltenyi Biotec, respectively). However, these antibodies failed to recognize Chinese hamster targets. 

According to your suggestion, we evaluated CD24/CD133 expression in CHK cells by qRT-PCR (Fig 3D). CD24 and CD133 were expressed in immortalized CHK-Q cells at least 4- and 19-fold higher than in primary cells, respectively. 

We added the related sentences in Page 11, 175, Page 17, Line 289 and Page 18, Line 307. 

(Page 11, Line 175)

For quantitative RT-PCR analysis, total RNA of cells was extracted and reverse-transcribed into cDNA using reverse-transcriptase and oligo-dT primers. cDNAs were mixed with corresponding primers (S1 Table) and quantitative PCR reagent (Thunderbird SYBR qPCR Mix, Toyobo) in accordance with the manufacturer’s instructions. PCR was performed using a quantitative PCR device (QuantStudio3, Applied Biosystems, Waltham, MA, USA) under the following conditions: 95°C for 1 min, followed by 45 cycles of amplification at 95°C for 15 s, 58°C for 15 s, and 72°C for 30 s, with a final cycle of amplification for melting curve analysis at 95°C for 30 s, 58°C for 30 s, and 95°C for 30 s. The fold change in the CD24 and CD133 specific transcripts relative to the glyceraldehyde-3-phosphate dehydrogenase (GAPDH) endogenous control gene was determined by the ΔΔCt method. The mRNA expression levels were expressed as mean values with standard deviations.

(Page 17, Line 289)

By quantitative RT-PCR analysis, the expression levels of CD24 and CD133 in immortalized CHK-Q cells were increased by at least 4- and 19-fold, respectively, compared with those in primary cells (Fig 3D).

(Page 18, Line 307)

(D) Expression analysis of kidney stem cell markers (CD24 and CD133) by quantitative RT-PCR. Data are expressed as mean ± SD (n = 3).

Comment 4:

The authors referred “The immortalization might be caused by increased gene expression fluctuation and activation of proto-oncogenes, suggesting that CHK cells are derived from renal stem cells.“ The analysis of the causes of spontaneous immortalization in this cell line is overly dependent on the results of DNA microarray. If the authors can proceed further study about the relation between proto-oncogenes and cell immortalization I believe the article will be very interesting.

Response 4: 

Thank you for the comment. However, the detailed relation between immortalization and proto-oncogene expression is still unclear and further study should be needed.

To Reviewer 2:

We would like to appreciate your comments that helped us to improve our manuscript. We revised the manuscript according to your comments.

Comment 1:

The authors detect the expression of CD24 and CD133, which are renal stem cell markers. Did the authors detect other renal stem cell marker genes? There is not enough evidence to suggest CHK-Q cells might be transformed from renal stem cells.

Response 1:

Relative expression profiles of renal stem cell marker genes from the DNA microarray data are summarized in Fig. S6. 

We added the following sentences in Page 17, Line 291.

(Page 17, Line 291)

The elevated expression of other renal stem cell marker genes such as paired box 2 (PAX2), paired box 8 (PAX8), homeobox B7 (HOXB7) and CBP/p300 interacting transactivator with Glu/Asp rich carboxy-terminal domain 1 (CITED1) [33,34] was detected in immortalized CHK cells (S6 Fig).

Comment 2:

The authors established a new immortalized kidney cell lines from Chinese hamster tissue. What is the advantage of these new immortalized kidney cell lines? What is the application of CHK-Q cells?

Response 2:

Thank you for this comment. We observed renal cyst-like structure formation in a gel sandwich culture of CHK-Q cells. Renal cyst-like structure formation has not been reported using other kidney-derived cell lines. We think that CHK-Q cells can be used to construct a renal cyst model for evaluating cell polarity and drug transport in kidney. We do not intend to include the results in this paper, but this will be reported in the future. 

Comment 3:

The chromosomes bear structural abnormality and undergo changes in structure and number during cell proliferation in CHK-Q cells. The cell line is unstable and may lose productivity over time in the manufacturing process. In the manuscript, the authors suggest the cell line could be used as a host for the production of biopharmaceuticals, alongside CHO cells and other kidney-derived cell lines. Did the authors detect productivity (for example: lentivirus package) over time in the lab?

Response 3:

Although CHK-Q cells have not been applied to lentivirus production, recombinant antibody genes have been introduced into CHK-Q cells. Recombinant CHK-Q cells showed stable production of the target protein, demonstrating that CHK-Q cells can be used as host cells for producing biopharmaceutical proteins. These data will be published in the future. 

Comment 4:

In lines 150-151: “Data were quantified using Feature Extraction software (Agilent Technologies) and normalized using R statistical processing software.” How do the authors normalize the DNA microarray data? Is raw intensity data for each experiment log10 transformed and then used for the calculation of Z scores?

Response 4: 

Data were quantified using Feature Extraction software (Agilent Technologies). The raw signal intensities of samples were log2-transformed and normalized by a quantile algorithm in the preprocessCore library package on Bioconductor software [28,29].

According to your suggestion, we modified the related sentences and added the following sentences in Page 10, Line 160, together with the references.

(Page 10, Line 160)

Data were quantified using Feature Extraction software (Agilent Technologies). The raw signal intensities of samples were normalized by a quantile algorithm in the preprocessCore library package on Bioconductor software [28,29].

(References)

28. Bolstad BM, Irizarry RA, Astrand M, Speed TP. A comparison of normalization methods for high density oligonucleotide array data based on variance and bias. Bioinformatics 2003;19:185-193.

29. Gentleman RC, Carey VJ, Bates DM, Bolstad B, Dettling M, Dudoit S, et al. Bioconductor: open software development for computational biology and bioinformatics. Genome Biol. 2004;5:R80.

Comment 5:

In lines 263-265: “(PCA) showed that the primary cells were different to CHK-Q cells and completely different to the CHO-K1 cells used as a control (Fig 3A). A similar tendency was observed in the heat map analysis (Fig 3B).” However, it is not easy to find this tendency in Fig 3B.

Response 5:

Hierarchical clustering using the Pearson's correlation distance/average linkage method was added on heatmap in Fig 3B. 

(Page 18, Line 305)

Hierarchical clustering was analyzed using the Pearson's correlation distance/average linkage method.

Comment 6:

These figures are low-resolution images. Figure legends maybe can separately from Results part on a page, at the end of the manuscript.

Response 6:

High resolution images can be downloaded from the journal homepage. The download link can be found in each figure. Regarding the preparation of figure legends, we followed the journal’s instructions.

---

## [Decision Letter · Decision Letter 1]

14 Mar 2022

Novel cell lines derived from Chinese hamster kidney tissue

PONE-D-22-00223R1

Dear Dr. Kamihira,

We’re pleased to inform you that your manuscript has been judged scientifically suitable for publication and will be formally accepted for publication once it meets all outstanding technical requirements.

Kind regards,

Wuqiang Zhu, MD, PhD

Academic Editor

PLOS ONE

Additional Editor Comments (optional):

Reviewers' comments:

Reviewer's Responses to Questions

**Comments to the Author**

1. If the authors have adequately addressed your comments raised in a previous round of review and you feel that this manuscript is now acceptable for publication, you may indicate that here to bypass the “Comments to the Author” section, enter your conflict of interest statement in the “Confidential to Editor” section, and submit your "Accept" recommendation.

Reviewer #1: All comments have been addressed

Reviewer #2: All comments have been addressed

2. Is the manuscript technically sound, and do the data support the conclusions?

Reviewer #1: Yes

Reviewer #2: Yes

3. Has the statistical analysis been performed appropriately and rigorously? 

Reviewer #1: Yes

Reviewer #2: Yes

4. Have the authors made all data underlying the findings in their manuscript fully available?

Reviewer #1: Yes

Reviewer #2: Yes

5. Is the manuscript presented in an intelligible fashion and written in standard English?

Reviewer #1: Yes

Reviewer #2: Yes

6. Review Comments to the Author

Reviewer #1: Although the data presented by the authors did not quite meet my expectations, the authors explained why and provided some additional experimental data to support their experimental results. It is hoped that there will be better experimental reagents in the future, otherwise the application of this cell line will be limited.

Reviewer #2: The authors have adequately addressed my comments, the manuscript has improved from its previous version.

7. PLOS authors have the option to publish the peer review history of their article (what does this mean?). If published, this will include your full peer review and any attached files.

Reviewer #1: No

Reviewer #2: No

---

## [Editor Report · Acceptance letter]

24 Mar 2022

PONE-D-22-00223R1 

Novel cell lines derived from Chinese hamster kidney tissue 

Dear Dr. Kamihira:

I'm pleased to inform you that your manuscript has been deemed suitable for publication in PLOS ONE. Congratulations! Your manuscript is now with our production department. 

Kind regards, 

on behalf of

Dr. Wuqiang Zhu 

Academic Editor

PLOS ONE